# Performativity, identity formation and professionalism: Ethnographic research to explore student experiences of clinical simulation training

Tanisha Jowsey[1]*, Lynne Petersen[2], Chris Mysko[3], Pauline Cooper-Ioelu[4], Pauline Herbst[3], Craig S. Webster[5], Andy Wearn[6], Dianne Marshall[7], Jane Torrie[6], Meng-Jiun Penny Lin[8], Peter Beaver[9], Johanne Egan[3], Kira Bacal[10], Anne O'Callaghan[11,12], Jennifer Weller[11,13]

1 Centre for Medical and Health Sciences Education, The University of Auckland, Auckland, New Zealand, 2 School of Pharmacy, The University of Auckland, Auckland, New Zealand, 3 Waitemata District Health Board, Auckland, New Zealand, 4 Learning and Teaching Unit, Faculty of Medical and Health Sciences, The University of Auckland, Auckland, New Zealand, 5 Centre for Medical and Health Sciences Education and Department of Anaesthesiology, The University of Auckland, Auckland, New Zealand, 6 The University of Auckland, Auckland, New Zealand, 7 School of Nursing, The University of Auckland, Auckland, New Zealand, 8 School of Education, The University of Auckland, Auckland, New Zealand, 9 Centre for Medical and Health Sciences Education, The University of Auckland, Auckland, New Zealand, 10 Medical Programme Directorate, The University of Auckland, Auckland, New Zealand, 11 Auckland City Hospital, Auckland, New Zealand, 12 Department of Psychological Medicine, The University of Auckland, Auckland, New Zealand, 13 Centre for Medical and Health Sciences Education, School of Medicine, The University of Auckland, Auckland, New Zealand

* t.jowsey@auckland.ac.nz

## Abstract

Developing professional identity is a vital part of health professionals' education. In Auckland four tertiary institutions have partnered to run an interprofessional simulation training course called Urgent and Immediate Patient Care Week (UIPCW) which is compulsory for Year Five medical, Year Four pharmacy, Year Three paramedicine and Year Three nursing students. We sought to understand student experiences of UIPCW and how those experiences informed student ideas about professional identity and their emergent practice as health professionals within multidisciplinary teams. In 2018, we commenced ethnographic research involving participant observation, field notes, interviews, photography and observational ethnographic film. A total of 115 students participated in this research. The emergent findings concern the potentially transformative learning opportunity presented within high fidelity multi-disciplinary simulations for students to develop their professional identity in relation to peers from other professions. Our work also exposes the heightened anxiety and stress which can be experienced by students in such interdisciplinary simulations. Student experience suggests this is due to a range of factors including students having to perform in front of peers and staff in such simulation scenarios when their own professional identity and capabilities are still in emergent stages. Staff-led simulation debriefs form a critical success factor for transformative learning to be able to occur in any such simulations so that students can reflect on, and move beyond, the emotion and uncertainty of such

**Data Availability Statement:** Consistent with our ethics approval, we cannot release data from this project. We did not seek consent from participants

to make the data set fully accessible. If readers wish to access the data they will need to make an application to the University of Auckland Human Participants Ethics Committee (humanethics@auckland.ac.nz) for data release.

**Funding:** This study was internally funded by the Faculty of Medical and Health Sciences at the University of Auckland (FDRF Grant: 3715764). Investigators who were named on this grant and who are authors on this paper are: TJ, JW, AW, JT, and AO. The funder had no role in the study design, data collection, analysis, decision to publish, or preparation of this manuscript.

**Competing interests:** The authors have declared that no competing interests exist.

experiences to develop future-focused concepts of professional identity and strategies to support effective interprofessional teamwork.

## Introduction

The use of simulation with undergraduate students within clinical training courses has recently been researched in terms of student preparedness to attend real-life ward calls[1], fidelity[2, 3], stress [4], patient management and error recognition [5], teamwork[1, 6, 7], and ethics of simulated death [8]. Research has also attended to the extent by which simulation training may be more effective than traditional clinical education [9, 10]. In their 2011 systematic literature review, McGaghie and colleagues note that while simulation-based medical education with deliberate practice is typically more expensive than traditional medical education, it is also more effective in some domains of learning, such as achieving specific clinical skill acquisition goals [9].

Student preparedness and skill acquisition are essential in clinical education to ensure workforce-ready graduates [11]. In Auckland, New Zealand, a simulation training course with undergraduate medical, nursing, pharmacy and paramedicine students has been underway since 2013, called Urgent and Immediate Patient Care Week (UIPCW) (see Box 1).

The present study was conducted to identify whether we were missing any important aspects of the student experience of UIPCW in relation to learning outcomes around preparedness and skill acquisition, professional identity and practice. We asked, what are student experiences of this simulation course and how do such experiences inform their workforce

---

### Box 1. Urgent and Immediate Patient Care Week—Prior research.

**Medical**: One year after experiencing UIPCW, when medical students became Trainee Interns (first year of clinical practice internship), students who had participated in the training were twice as likely to take an active role in ward calls than peers who had not attended the training (1). Student evaluations showed that medical students valued learning about the scopes of practice of nurses and pharmacists in terms of leadership, roles, and responsibilities.

**Pharmacy**: In a further study (7), pharmacy student evaluations suggested that their key learning areas were prioritisation of care, systematic assessment of patients, appreciating scopes of practice and communication strategies.

**Nursing**: Nursing student evaluations (unpublished) suggested they had developed effective ways to share information with other members of the care team in a timely fashion. These different foci of students from the various disciplinary groups reflected, to a large extent, their professional roles and responsibilities within urgent and immediate patient care. In the workplace, for example, nurses are often first on the scene when a patient deteriorates. Thus, nurses in UIPCW were encouraged to focus on serial assessment and management of the patient, including hand over of relevant information to other members of the healthcare team.

readiness? How do these student experiences inform their ideas about professional identity and future scopes of practice?

## Methods

A critical ethnographic approach underpins this methodology in order to illuminate the values, meanings and scopes of practice informing undergraduate health student experiences [12].

### Ethics

Ethics approval for this research, reference number 020759, was obtained from the University of Auckland Human Participants Ethics Committee. In the consent forms, participants were asked to tick a box for consenting to participate in the research and to tick a second box if they also consented to being filmed. The film consent details followed film industry-standard consent protocol. All photos included in this manuscript identify people who consented to participate in the research and to be photographed and for the photographs to be included in research outputs.

### Setting

Urgent and Immediate Care Week (UIPCW) is an initiative that arose from strong evidence around effectiveness of simulation and situated practice [6, 10, 13–15]. UIPCW is a compulsory four day course for final year medical, nursing and pharmacy students from the University of Auckland (UOA), Auckland University of Technology (AUT), Manukau Institution of Technology (MIT) and Unitec Institute of Technology (UIT). UIPCW runs seven times across the academic year as a four-day (consecutive) course. Throughout the four days students are given the opportunity to work with students from other disciplines on clinical problems. Students are led through a series of in-class learning activities that contribute to their skill acquisition and, in part, prepare them for high fidelity simulation scenarios. These scenarios are drawn from a number of different settings including the community and ward setting. Students are challenged to reflect on their professional roles, communication and teamwork alongside other students and faculty from a range of specialties and disciplines. They are encouraged to reflect on their own performance within their own scope of practice, and in relation to other health professionals' scopes of practice.

### The evolution of UIPCW

In 2013 The University of Auckland developed and piloted an interprofessional simulation course called WardSim for Year Four pharmacy students, Year Three nursing students, and Year Five medical students. It was made compulsory for students in 2014. In 2017, collaboration with Auckland University of Technology brought paramedicine into the course, for whom participation is not compulsory. Nursing students and faculty from the Manukau Institute of Technology (2017) and Unitec Institute of Technology (2018) joined University of Auckland nursing faculty and students, creating UIPCW in its current form.

### UIPCW goals

Overarching goals of UIPCW are to offer students immersive and authentic learning opportunities around interprofessional communication, leadership and teamwork skills. Specific objectives for all students include: systematic approach to clinical assessment, interprofessional teamwork and role clarity, recognition of knowledge gaps and how to request help, effective

> ## Box 2. Urgent and Immediate Patient Care Week—Programme components.
>
> Across the four days the focus shifts from community care (day 1) to palliative care (day 2) and acute inpatient care (days 3–4). Medical students are present on all four days while pharmacy students are present for days one, two, and four. Paramedic students are only present for the first two days, and nursing students are present for the final two days. Students are allocated to interprofessional groups of 8–12 students. Groups are maintained throughout the four-day course. Groups rotate between interactive class-room-based activities and simulated scenarios.
>
> Simulation training
>
> A smaller mixed professional team of students (with at least one from each profession present) participates in each simulation; all students are involved in at least one scenario and otherwise observe their peers. Structured debriefs follow each simulation to explore reactions, 'unpicking' actual events with a focus on how learning transfers to the real workplace. Facilitators strive to cultivate a safe and positive atmosphere during debriefing with emphasis on transfer of learning of future-focused principles and practices.

use of communication tools—for example ISBAR (identification, situation, background, assessment, recommendation) and PACE (probe, alert, challenge, emergency) for graded assertiveness in speaking up about concerns [16] in dynamic, urgent health settings and situations. For UIPCW programme components see Box 2.

## Study design and observation

Using an ethnographic approach [17–20], we closely explored social lived experience and communication of staff and students during UIPCW (cycles three and four in May 2018) via observation, field notes, photographs, observational film and interviews with research participants. Often, the ethnographer is 'immersed' in the research field over a long period of time (generally 3–24 months) in order to get an in-depth knowledge and understanding of the research participants in their environment [21]. Instead of using this traditional ethnographic method we did a 'focused ethnography'[22], whereby we had four researchers immersed in the research field over a short period of time (four days for each cycle). The lead investigator on this project (first author) had contributed to teaching and research for UIPCW for three years prior to undertaking this ethnographic research. So although the method is 'time compressed ethnography' it is informed by long-term familiarity and immersion with the course. Our ethnographic approach was observational. The research observers each carried an A5-sized notebook for field notes and on average, each researcher took 55 to 70 pages of notes per cycle (a mean of 16 pages per day). Researchers observed staff and student participants and carried out short (1–5 minute) unstructured interviews during course breaks. These informed their field notes, which were then synthesised to underpin our research findings. None of the teaching faculty in this study were part of the observer team.

During cycle four a film crew of two people from the University of Auckland's Media Productions Unit was present. They brought six cameras, which were used to capture different aspects of student experience. As part of our reflexive strategy, two members of the research team monitored the effects of the film crew on research participants. Of the participants who

gave informed consent to be filmed, the research team chose students to focus on. In medium-length documentaries it is common to focus on the experiences of up to six people. Over 100 hours of footage was taken during cycle four and was edited into a 26-minute ethnographic film called *Prepared to Care* [23]. Footage was purposively selected for the film based on its relevance to the key findings and consistent with ethnographic film techniques. Editing was conducted by the first author and a professional film editor (Richard Smith of UoA Media Productions) using Adobe Premiere Pro software.

## Analysis

Ethnographic analysis entails analysis of data in concert with social theory [24]. We undertook an iterative process of analysis [25]. Three members of the research team [TJ, PL, P-CI] collated, summarised and thematically coded the observational written data (not including film data) by hand [25]. Similar codes were grouped together to create higher order concepts [26]. We checked these concepts back against the data regularly to ensure that interpretation was consistent. We read and re-read the entire dataset to ensure data immersion. At the same time as this analysis was underway, one member of the research team [TJ] was editing the ethnographic film output of this study, which added an audio-visual mode of data immersion. The film data was analysed by hand and film analysis/editing notes were made in Microsoft Word software. This in-depth process of data immersion and coding allowed the team to agree on emergent themes, which formed the basis for the ethnographic engagement with existing social theory. The social theory that we identified as most relevant concerns performance and performativity, which we now detail.

## Theoretical context: Performance and performativity

The aspect of performance that seems to be barely accounted for in health professions education literature is that of performance as acting and presenting particular aspects of the self in an intentional and practised way; i.e. as performativity [27–29]. In this paper we look closely at *performativity* as intrinsic to the student experience.

Performance is both an *act of presenting* and an *evaluative process*. In healthcare simulation discourse, performance is often described in terms of how well equipment performs, how a course increases technical skill acquisition and mastery, and team performance [27–29]. Performance in these senses is intrinsically linked to evaluative notions of what it means to competently perform a skill or other professional aspects of clinical work. However, performance is also acting and presenting particular aspects of the self in an intentional and practiced way; as performativity. The term performativity has, unsurprisingly, been used a lot in theatrical performance studies. As we grappled with making sense of our data we realised that performativity theory provides a useful and appropriate lens through which we can further understand healthcare learning through simulation. Performativity is the convergence of a dynamic set of forces that play out in performances. In contrast to performance as an evaluative process, performativity not only looks to the end product—competence—but also explores language, structure, behaviours and context to elicit meaning during the process of performing. Performativity includes both theatrical components (actions students take because they perceive an expectation that they do so) and improvisational components. In healthcare, theatrical components include such things as jargon; name tags, white coats, and uniforms [30]. Improvisational components are what the performer does, and this is conditioned by their prior experiences (including experiences of hierarchy and agency) [30].

Performativity is always motivated to achieve the goals at hand by, and imbued with the risk of failure [31]. In the context of any performance, what constitutes risk is intrinsically

open to interpretation, negotiation, and contestation, as the performers do not perform alone but rather with others. There is also cultural value attached to risk taking; people do not exist in a social or cultural vacuum and decisions about risk are socially embedded and shaped by culturally based notions of the world, what the world consists of, and how it works [31]. As Grant writes,

> "The decision offers itself to the performer. Failure to take or recognise the decision is what is at risk in the performance. In a performative event, that's the performative moment, the performer always hovers at the cusp of possibility, borne and buffeted by the history of their own training, the script or score, the cultural tradition, the specific situation, the actions of other performers and the lure of the ultimately unknown possibility which beckons"[32: 130].

What Grant is saying here is that the performer—the person in the moment of having to make a decision and act upon it—experiences *risk* in terms of that very decision they are making. Again, that decision is based on the performer's history and experiences leading up to that moment. Performativity, therefore, can be seen in these individual, and often rapid moments of *decision making*. In the simulation setting, people are often required to quickly make decisions and act on them, where risks are either embraced, shied away from, ignored, or abandoned. In this paper, we are conceptualising risk in terms of the way a person perceives their personal exposure or the exposure of their patient to danger, harm, or loss (including loss of face). In the case of personal risk, this may involve risk of being seen to make an error and being judged socially—by teachers and peers—on performance.

Performativity deals in the subtleties of the performance; the pauses, gestures and as much in what is unsaid, as well as what is said. It has been argued that anthropological analysis should encompass not only performance but also intentionality [33]. Even though only the enacted or performed is visible, anthropology must study the relationship between what is performed and the underlying cognitive models and social structures that underpin each performance. Using this framework of analysis, we argue that performativity is intimately linked to notions of identity and personhood, which we explore in the findings in terms of how students developed their own notions of roles and team care, as framed by their emergent notions of their roles and identities within health teams.

## Results

### Participant characteristics

A total of 115/153 (75% response rate) students participated in this research, including 58 of 73 and 54 of 80 students from cycles three and four respectively (Table 1). Eighty-three staff were invited to participate and of these 78 consented (94% consent rate).

Faculty members were drawn from the medical, nursing, pharmacy and paramedicine programmes. Most members had an interest in education and had completed at least a half-day formal training session in scenario-based learning including debriefing. The typical ratio of faculty to students was 1:5 and of simulation technicians to students 1:20.

The ethnographic findings presented here detail student experiences of UIPCW in terms of learning through performing, professionalism, and debriefs.

### Learning through performing

Day one of UIPCW in both cycles began in a similar way. Students arrived five minutes before the start time of 8:00 am, entered the room in a queue, collected their name badges and

**Table 1. Characteristics of students and staff who consented to be observed.**

| Research participants | Cycle Three | | Cycle Four | |
| --- | --- | --- | --- | --- |
| | May 8–11 2018 | | May 22–25 2018 (filming) | |
| | N = 73 | | N = 80 | |
| **Students** | Attended | Consented to be observed | Attended | Consented to be observed |
| Medical | 38 | 31 | 39 | 22 |
| Nursing | 16 | 11 | 22 | 16 |
| Paramedicine | 8 | 8 | 8 | 7 |
| Pharmacy | 11 | 8 | 11 | 9 |
| **Subtotal students** | **73** | **58** | **80** | **54** |
| **Staff** | | | | |
| Medical | 11 | 11 | 13 | 13 |
| Nursing | 4 | 3 | 5 | 3 |
| Paramedicine | 3 | 3 | 5 | 5 |
| Pharmacy | 7 | 7 | 7 | 7 |
| Palliative | 3 | 3 | 3 | 3 |
| Radiographer | 1 | 1 | 0 | 0 |
| Medical educator | 2 | 2 | 0 | 0 |
| Simulation Technician | 3 | 3 | 4 | 4 |
| Time keeper/researcher | 2 | 0 | 2 | 2 |
| Actor patient | 4 | 4 | 4 | 4 |
| **Subtotal (staff members)** | **40** | **37** | **43** | **41** |

paperwork, completed the research consent forms and sat down. There were quiet conversations and as students joined their assigned table they continued to speak to peers from within their profession. A clinical staff member began an introduction to UIPCW and the room immediately fell quiet except for her voice. Students were oriented to the dense minute-by-minute timetable. Then the first group activity began. Students then introduced themselves to their group members and visibly relaxed a little—yawning, joking, stretching back in chairs—as they began to work together. The room remained relatively quiet, which was a pattern that students kept at the beginning of each day throughout UIPCW. Students expressed feeling nervous about simulations to each other and we noted they seemed to have varying levels of confidence. In the following section we discuss nerves and confidence in relation to performativity, risk and professionalism.

The first simulation was scheduled to begin after morning tea break on day one. During the break students separated from their assigned interprofessional groups and joined peers from their own professions. There was a general feeling of tension and they almost unanimously reported feeling "nervous". They were nervous about the upcoming simulation scenario; how they would perform, whether they would make mistakes or have a stress-induced "mind blank", and what their observing peers would think of them if they performed badly. Some students said they experienced sweating, increased anxiety and in two cases an urgent need to use the bathroom. Students desired to perform, and be seen as performing, appropriately within the disciplinary boundaries of their profession specifically, and of healthcare more broadly.

At the beginning of the first simulation, students were separated into two sub-groups: the observers and the simulation scenario. Observers watched from behind one-way glass in an observation room while the others waited in the hall for staff to signal them to enter the simulation room (Figs 1–3). Once they entered the room students could not see their observers but they knew they were being watched—closely—and by several people.

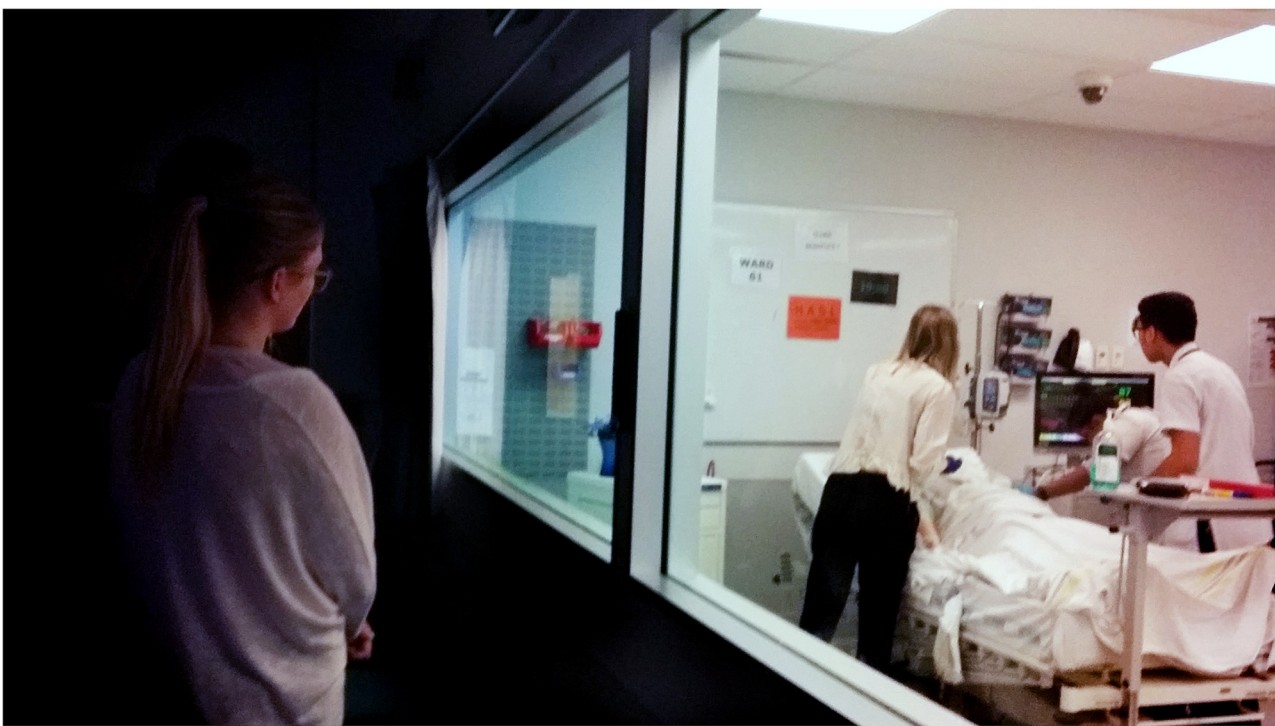

**Fig 1. Students in simulation scenario with student peers observing through one-way glass.** Image taken with permission, by the first author, May 2018.

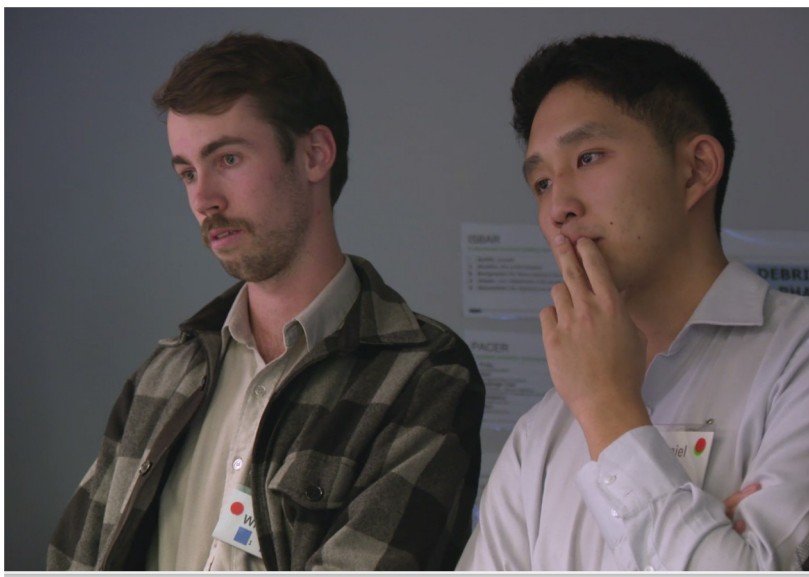

**Fig 2. Students observe their peers through one-way glass.** Image taken with permission, by the first author, May 2018.

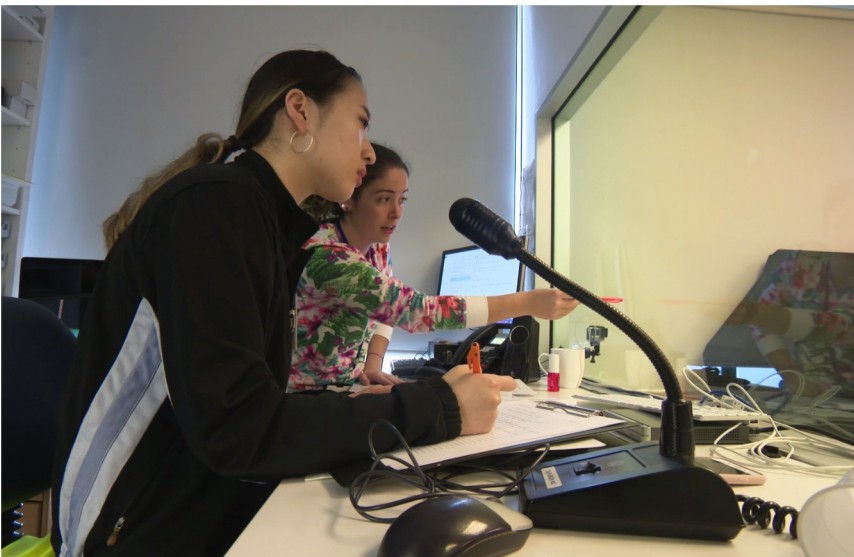

**Fig 3. Staff observe students through one-way glass.** Image taken with permission, by the first author, May 2018.

Following the simulation, all students from the group sat in a circle in the debrief room joined by two or three teaching staff. First, students reported their feelings. The most frequent reactions described were "nervous", "overwhelmed", "anxious", "uncertain" and "flustered." Students wondered "am I going down the right track?" and reflected, "[I'm] glad to get it over and done with." Observing students often said things like "I felt nervous for them" and "I was glad it wasn't me in there."

### Reflecting in action about performativity

Following simulations in structured debriefs, facilitators assisted students to unpack their strong feelings along with what happened during the experience with an emphasis on how those in the simulation performed. Performativity was inherent in statements such as "it's weird because you know everyone's watching but you can't see them." During debriefs students demonstrated strong desires to establish what they should do when faced with a similar real-life situation. Common learning points were "start with ABC", "tell someone whether or not I can do something", "regroup and invite ideas", and "keep your medical brain straight." Many medical students talked about "wanting facts" and did not focus so much on feelings associated with high pressure to perform. This sense-making—such as clarifying protocol and procedure—reflected current professional discourse and embedded ideas of professionalism, as well as idealised notions of what students understood to be appropriate ways of acting like a health professional. Following the debrief students looked much calmer, looked less tense, and often smiled. The debrief had either reassured students that they were performing appropriately according to their discipline, or they were given modifications to their professional script. One student commented, "I felt relief [after the problem was solved], a good experience."

Another aspect of the simulation scenarios that informed student performativity was the involvement of actor 'patients' and/or computerised manikins. The strengths and limitations of actors and manikins for simulation learning are widely understood [13]. For student participants the strength of actor 'patients' was in their capacity to bring into sharp focus complex and confronting emotions—such as when the actor 'patient' asked whether their loved one

was still alive, or when they burst into tears while discussing their advance care plan options, or in instances when they became angry about the quality of their care. In these moments fidelity was high and students reported feeling overwhelmed or immediately compassionate toward the 'patient'.

Nursing students stood out as particularly strong in their compassion and communication skills with manikin 'patients.' This may be due, in part, to their larger number of actual clinical hours of experience working with patients at a similar point in their undergraduate degree structure in comparison to the other undergraduate health students. The nursing students spoke clearly and kindly with the 'patient', kept them informed, asked permission, used the 'patient's' name, and touched their arm to provide comfort or reassurance. These elements all contributed to the success of their performance and implied perhaps, a more 'developed' sense of professional identity and capability formation or perhaps a professional identity revolving around a clear sense of patient-centred focus. In general, the medical students appeared to struggle more than other disciplinary groups to communicate compassionately with 'patients.' One nervous student, for instance, told the patient that they had suffered a heart attack and followed this cheerfully with "you are in the best place for it." Their intention was to provide reassurance but the patient was not reassured. In another simulation a medical student spoke with a palliative patient whose voice became weak and quiet. The student was standing at the end of the bed. He gradually increased his volume as he spoke and gestured for the patient to speak louder (which the patient did not do). During the debrief the group discussed whether moving closer to the patient might have been a more effective strategy to demonstrate compassion and promote effective communication.

## Responding to professional uncertainty: Confidence and anxiety

Some students' observed actions (or lack of actions) within the simulations can be understood as a response to their less solidly formed identity and/or their less developed experience in applying their emerging professional capabilities. Paramedicine students, for example, had experienced considerably more simulation training than their peers had prior to the UIPCW simulation course. Not surprisingly, paramedicine students were more confident and vocal about their abilities than other students. This confidence is an indicator that they were more familiar with the routines, requirements, and skills associated with a successful 'performance' and therefore, did not experience anxiety as often as other students. When reflecting on their performance in simulations paramedicine students said to each other, "we are quite good at communicating", "we even involve family members" and "we advocate for our patients." Their confidence was evident during simulations, and also shone throughout classroom-based activities where they sat more up-right and engaged in whole class discussions more than their peers did. Medical students noticed the confidence and capability of paramedicine students in simulations. On many occasions students made reference to the confidence and competence of paramedicine students to make a case for greater and earlier inclusion of simulation training in their own curriculum.

The flipside to student personal and shared confidence, is their perceptions of, and responses to, the personal uncertainty about how to perform adequately, worries about missing something obvious, failing to correctly diagnose and treat the simulated patient, or appearing unprofessional. When asked what their biggest concerns were, medical students responded with such comments as "that I might make a mistake but it's a good environment to learn in." How students respond to uncertainty is informed by their prior experience. In this case, with prior experience of simulation training, of working with students from other professions, and of caring for patients. Responding to uncertainty has elsewhere been discussed in terms of

perceptions of risk and responses to it [31, 34–37]. We suggest that prior student experiences and their *discipline*–as social status—also significantly informed their uncertainty (perceptions of risk) and confidence. This also influenced the discourse they projected and promoted amongst their (inter-) professional peers.

## Professionalism enacted

Healthcare professionalism is a demonstration of essential values enacted through personal professional behaviours and relationships: caring for and respecting patients, acting honestly and ethically, working in partnership with patients and colleagues, and accepting the obligation to maintain and improve standards [38, 39]. Current professionalism discourses tend to cover four spaces: technical (scale); communicative (conversation); improvement (consensus conference), and critical (process) [40, 41]. Professionalism is demonstrated in each of these spaces. For example, in technical and communicative ways, professionals might wear a uniform, or display symbols of their profession in particular established ways; a stethoscope around their neck or an upside-down timepiece clipped to their breast pocket. Throughout UIPCW students from their various professions demonstrated what professionalism meant to them. At a basic level this was largely embodied and symbolic (i.e. uniforms) [42]. The personal characteristics and demonstrations of professionalism—such as moral duty of care, integrity, patient-centred approaches, and so forth—were embodied through participants' actions and voice.

Students from different professions were clearly at different stages of their professional identity formation [43, 44] as evidenced by how comfortable they were within both the simulation scenario itself as well as in the post-simulation debrief. This can be understood by the varied experiences in their respective training courses prior to UIPCW and for them as uniquely emergent professionals.

Students had various ways of demonstrating and performing care through practice within the expected boundaries of their professional role. Nursing and paramedicine students demonstrated 'care by doing' and medical students demonstrated 'care by enquiring.' Pharmacy and nursing students oscillated between 'care by enquiring' and 'care by examining the clinical notes' at the end of the bed, which is a legitimate caring role; though some students often hovered at the end of the bed long after they had ceased engaging with the clinical notes and documentation. In simulations, once a medical student arrived, it was observed that pharmacy students often seemed unsure about whether they were still needed and in what capacity. Student motivations to be interpreted as 'performing in a professional way' was evident through such actions as finding an appropriate place to stand or sit in the simulation room. A paramedicine student said that prior to entering a simulation scenario she told herself "Don't be a dick. Just be professional. Go in there and get the job done." For her, professionalism meant getting 'the job done' and not letting her nervousness impair her performance.

## Debriefs as future-focused reflection on professional performativity

The purpose of debriefs is to provide a constructive safe space in which learners can make sense of and learn from what happened during the simulation scenario. Facilitators were clearly cognisant of this and they worked hard to support student learning in debriefs by communicating effectively, directing conversation to learning points, clarifying technical points as needed, and drawing each learner into the conversation. During debriefs students actively sought to make sense of what had happened during the scenario and how successfully they had performed. People who had played a leading role at various points during the scenario discussed their thoughts and actions they had had. Observing students reassured them, saying

"you did really well" and "I thought it was good that you [another student] pressed for help." Staff played a critical role in directing debriefing discussions towards key areas of future successful professional team-based 'performance' including role clarity and effective team communication. Debriefs informed student ideas about their own role and identity, and how to contribute to effective team care in future team-based situations.

## Discussion

In previous research we identified key aspects of the student experience in such simulations in terms of critical learning about structured communication, leadership, scopes of practice, and teamwork. Findings from the present ethnographic study align with these findings and make a case for UIPCW being a fundamental space in which students form and reform their professional identities. They undertake this important transformative work through extending their knowledge and skills, by attempting to perform as their desired future professional selves and through skilled debriefed reflections around implications of note for their future identity and practice.

Medicine and health science qualifications transform people into being responsible for people's lives. Students, while engaging in the 'performing' of their simulation learning experiences, were at the same time acting as placeholders for their future selves as healthcare professionals. It is therefore not surprising that students related their performativity to their professional status; a good performance indicated socially that they were morally responsible and could be trusted with patient health. Similarly healthcare involves risk and a very real danger to people's lives—a good performance shows that students understand their part in managing this within their professional roles.

The findings of this study raise important issues concerning performativity, professional identity formation in team settings and the development of professionalism for students learning interprofessionally in simulation contexts. Barad describes performativity as "enactment of boundaries—that always entails constitutive exclusions and therefore requisite questions of accountability"[45: 803]. We suggest that professionalism also entails such enactment, which is inherently difficult to navigate.

UIPCW and other equivalent high-fidelity training experiences may result in an emotive and therefore remarkable occasion in the formation of one's professional identity. This type of simulation experience seems to offer a unique opportunity for students to mirror as closely as possible essential professional practices, and consequently, for them to gain insights central in their professional identity formation. Based on theory generated by Pratt et al. [46] we suggest those lacking confidence in scenarios may have been experiencing a significant disparity between their self-identity and their professional 'actual work' identity. Previously-developed identities may have been required to 'splint' a weak emerging identity [46]. The high fidelity and social nature of interprofessional simulation supported students to move towards identity enrichment via experience and reflection; thus leading towards incremental self-awareness and understanding of what professionalism meant to them personally.

Finally, the importance of well enacted debriefs on student learning and identity formation cannot be overstated. In this study, debriefs stood out as central elements in the course through which students undertook much of the reflective 'work' that we have outlined in the discussion above. Other literature has identified effective feedback [47]–such as through debriefs—following simulation as imperative to optimising learning [48–50]. We would add that it is equally imperative to informing identity formation. Skilled debriefing can enable reflection on and 'moving beyond' the emotion of simulation experiences. As such, it can support learners through vulnerable spaces towards solidifying their ideas about professional scopes of practice,

how to work effectively with people from other professions, and the provision of safe and ethical care. Skilled debriefing faculty staff are essential to such outcomes.

## Limitations

This study is limited in that it only reports on the experiences of one group of students, who were a small sub-set of the respective disciplines' undergraduate students, in a very specific course at a single point in time. We cannot speak to the generalisability of participant experiences beyond this group, setting or time. Several authors had prior immersive experience with UIPCW as faculty, and this may have informed their biases and views of the data. However, three of the ethnographic observers had no prior experience of UIPCW, so this likely minimises such potential influence over findings reported here.

It is difficult to know how much the presence of a film crew informed student experiences of UIPCW. In cycle four, two stationary cameras were set up in one simulation room. A cameraman walked through the room with a motion camera on his shoulder and the director held the attached boom microphone over students' heads when the camera was pointing at them. During one of the simulations the actor 'patient' had a mounted Go-Pro camera on her forehead. There was also a camera technician on a computer hidden in the corner of the room behind a sheet, and two inbuilt cameras in the ceiling. Students had to overcome the limitations of simulation at the same time as ignore the multiple cameras and people that detracted from its realism. For the majority of the time, students seemed engaged in the learning simulation without apparent distraction or impact by the filming process.

## Conclusions

Clinical simulation training, such as UIPCW, creates an environment through which learners—in this case undergraduate pharmacy, medical, nursing, and paramedicine students—practice and perform. Through performativity in simulation scenarios and identity work in debriefs, learners develop their personal and emergent professional identities. This identity work is as important as the knowledge and skill development that also occurs. We recommend simulation training as an avenue for supporting the long and complex journey from novice to healthcare professional capable of contributing effectively to multidisciplinary team-based care. The importance of skilled debriefing to enable reflection on the emotion of such experiences forms a critical success factor for any such simulations.

## Acknowledgments

The authors thank all students and staff who participated in this study.

## Author Contributions

**Conceptualization:** Tanisha Jowsey, Lynne Petersen, Chris Mysko, Craig S. Webster, Andy Wearn, Dianne Marshall, Jane Torrie, Johanne Egan, Kira Bacal, Anne O'Callaghan, Jennifer Weller.

**Data curation:** Tanisha Jowsey, Pauline Cooper-Ioelu, Meng-Jiun Penny Lin, Peter Beaver.

**Formal analysis:** Tanisha Jowsey, Lynne Petersen, Chris Mysko, Pauline Cooper-Ioelu, Craig S. Webster, Andy Wearn, Dianne Marshall, Meng-Jiun Penny Lin, Peter Beaver, Johanne Egan, Kira Bacal, Anne O'Callaghan, Jennifer Weller.

**Funding acquisition:** Tanisha Jowsey, Lynne Petersen, Craig S. Webster, Andy Wearn, Dianne Marshall, Jane Torrie, Johanne Egan, Kira Bacal, Anne O'Callaghan, Jennifer Weller.

**Investigation:** Tanisha Jowsey, Pauline Cooper-Ioelu, Jane Torrie, Meng-Jiun Penny Lin, Johanne Egan.

**Methodology:** Tanisha Jowsey, Pauline Cooper-Ioelu, Pauline Herbst, Meng-Jiun Penny Lin.

**Project administration:** Tanisha Jowsey.

**Resources:** Tanisha Jowsey.

**Software:** Tanisha Jowsey, Pauline Cooper-Ioelu.

**Supervision:** Tanisha Jowsey.

**Validation:** Pauline Herbst.

**Writing – original draft:** Tanisha Jowsey, Lynne Petersen, Pauline Cooper-Ioelu, Craig S. Webster, Andy Wearn, Dianne Marshall, Jane Torrie, Meng-Jiun Penny Lin, Peter Beaver, Johanne Egan, Kira Bacal, Anne O'Callaghan, Jennifer Weller.

**Writing – review & editing:** Tanisha Jowsey, Lynne Petersen, Chris Mysko, Pauline Cooper-Ioelu, Pauline Herbst, Craig S. Webster, Andy Wearn, Dianne Marshall, Jane Torrie, Meng-Jiun Penny Lin, Johanne Egan, Kira Bacal, Anne O'Callaghan, Jennifer Weller.

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
