## [Decision Letter · Decision Letter 0]

15 Apr 2020

PONE-D-19-27048

Performativity, identity formation and professionalism: ethnographic research to explore student experiences of clinical simulation training

PLOS ONE

Dear Dr Jowsey,

Thank you for submitting your manuscript to PLOS ONE. After careful consideration, we feel that it has merit but does not fully meet PLOS ONE’s publication criteria as it currently stands. Therefore, we invite you to submit a revised version of the manuscript that addresses the points raised during the review process.

We would appreciate receiving your revised manuscript by May 30 2020 11:59PM. To enhance the reproducibility of your results, we recommend that if applicable you deposit your laboratory protocols in protocols.io, where a protocol can be assigned its own identifier (DOI) such that it can be cited independently in the future. For instructions see: http://journals.plos.org/plosone/s/submission-guidelines#loc-laboratory-protocols

We look forward to receiving your revised manuscript.

Kind regards,

Janhavi Ajit Vaingankar

Academic Editor

PLOS ONE

Journal Requirements:

2. To meet our criteria on data reproducibility, please ensure that the program has been adequately described (in particular, please consider reporting in more details the scenarios included in the simulation training.

3. We note that Figures 1, 2 and 3 includes images of participants in the study.

5. Your ethics statement must appear in the Methods section of your manuscript. If your ethics statement is written in any section besides the Methods, please move it to the Methods section and delete it from any other section. Please also ensure that your ethics statement is included in your manuscript, as the ethics section of your online submission will not be published alongside your manuscript.

Reviewers' comments:

Reviewer's Responses to Questions

**Comments to the Author**

1. Is the manuscript technically sound, and do the data support the conclusions?

Reviewer #1: Yes

Reviewer #2: Yes

2. Has the statistical analysis been performed appropriately and rigorously? 

Reviewer #1: N/A

Reviewer #2: N/A

3. Have the authors made all data underlying the findings in their manuscript fully available?

Reviewer #1: Yes

Reviewer #2: No

4. Is the manuscript presented in an intelligible fashion and written in standard English?

Reviewer #1: Yes

Reviewer #2: Yes

5. Review Comments to the Author

Reviewer #1: I have thoroughly enjoyed reading this well written manuscript. The use of focused ethnography is highly valuable in healthcare education and practice. I congratulate the authors conducting on such a fantastic piece of research. I have only four very minor points for consideration to the methods section:

Line 223: Focused ethnography is a valid method. Perhaps it is worth adding how the 26-minute content was selected from the overall 100 hours. What was the focus of the editing? This would add a little more transparency.

Line 228: Could you clarify which data this refers to - by hand as I initially presumed this excluded the video data? How was the video data coded? Did you use specific software, or simply make handwritten notes?

Line 231: Consider clarifying what constitutes the data set?

Lines 457-458: The word clearly is used twice in this sentence. Consider revision.

Reviewer #2: I am not an expert in ethnography or qualitative methods generally. This created several points of confusion and doubt on my part due to that ignorance but also due to the use of technical language and jargon in reference to the methods and the theoretical framework. Please read the paper for areas you could simplify and clarify. I will point out some that I encountered. The quote by Grant on page 10 was particularly dense and confusing.

Ethnography would be an appropriate methodology for this situation. The use of multiple observers seems a good choice and the brevity of the experience makes it feasible as well. I remain confused about the purpose of the filming and the ethnographic video that resulted. If this was a planned part of the study, please describe how it was intended to contribute and where the observations about the use of the film are reported and incorporated into the discussion and conclusions.

Similarly, it would be useful to have more detail about the simulation itself, particularly the nature and content o the simulations.

Performativity theory is appropriate for use in this study but I would like the authors to explain why they selected this framework - is it novel? Previously used by these investigators? Does it have a particular strength or nuance that helps this study? I found the sentence on page 10 lines 253-254 rather opaque - please clarify what you intend by theatirical and improvisational components. An example would be helpful. Similarly, please expand and illustrate your definition of risk. There are many ways to talk about and conceptualize risk, but I am unsure of the specifics of what you are using. Also, what do you mean by "fleeting moments of decision making (p 11 line 272)? please explain and illustrate in the context of the UIPCW.

The three figures mentioned on page 13 were absent from my version of the manuscript.

6. PLOS authors have the option to publish the peer review history of their article (what does this mean?). If published, this will include your full peer review and any attached files.

Reviewer #1: Yes: Suzanne Gough

Reviewer #2: No

---

## [Author Response · Author response to Decision Letter 0]

19 May 2020

Dear Editor,

RE: Performativity, identity formation and professionalism: ethnographic research to explore student experiences of clinical simulation training

Thank you for the opportunity to make the minor revisions requested for this manuscript. Following are our responses to the reviewer comments.

COMMENTS and RESPONSES

Reviewer #1: I have thoroughly enjoyed reading this well written manuscript. The use of focused ethnography is highly valuable in healthcare education and practice. I congratulate the authors conducting on such a fantastic piece of research. 

Thank you for the kind words

I have only four very minor points for consideration to the methods section:

Line 223: Focused ethnography is a valid method. Perhaps it is worth adding how the 26-minute content was selected from the overall 100 hours. What was the focus of the editing? This would add a little more transparency.

Change made in text to provide this information (line 223ff): Footage was purposively selected for the film based on its relevance to the key findings and consistent with ethnographic film techniques. Editing was conducted by the first author and a professional film editor (Richard Smith of UoA Media Productions) using Adobe Premiere Pro software.

Line 228: Could you clarify which data this refers to - by hand as I initially presumed this excluded the video data? How was the video data coded? Did you use specific software, or simply make handwritten notes?

Change made to clarify (lines 231-239) modes of analysis for the written and film data

…coded the observational written data (not including film data) by hand.

… The film data was analysed by hand and film analysis/editing notes were made in Microsoft Word software.

Line 231: Consider clarifying what constitutes the data set? Now clarified, as above.

Lines 457-458: The word clearly is used twice in this sentence. Consider revision.

Change made, second word ‘clearly’ replaced with “effectively”

Many thanks to reviewer one for their time in reading and reviewing with these helpful comments.

Reviewer #2: I am not an expert in ethnography or qualitative methods generally. This created several points of confusion and doubt on my part due to that ignorance but also due to the use of technical language and jargon in reference to the methods and the theoretical framework. Please read the paper for areas you could simplify and clarify. I will point out some that I encountered. The quote by Grant on page 10 was particularly dense and confusing.

Change made. While Grant’s quote is dense, we see it as being useful to explaining the complexity of performance as risk. It is a complex issue. It is difficult to say it simply. We have tried to say it simply in our text surrounding Grant’s quote. After careful consideration we have decided to keep Grant’s quote and to explain it more simply as well [lines 277-280] What Grant is saying here is that the performer – the person in the moment of having to make a decision and act upon it – experiences risk in terms of that very decision they are making. Further, that decision is based on the performer’s history and experiences leading up to that moment.

Ethnography would be an appropriate methodology for this situation. The use of multiple observers seems a good choice and the brevity of the experience makes it feasible as well. 

Thank you for your support

I remain confused about the purpose of the filming and the ethnographic video that resulted. If this was a planned part of the study, please describe how it was intended to contribute and where the observations about the use of the film are reported and incorporated into the discussion and conclusions.

Similarly, it would be useful to have more detail about the simulation itself, particularly the nature and content o the simulations.

No change made. The reviewer is asking for a more evaluative approach to the paper than an ethnographic one. The key elements of the course needed to understand the ethnographic approach are described (box 1 and 2), and that, in fact, one of the important purposes of the film is to offer the thickest description possible (as per Clifford Geertz) and experiential summary of the course itself (which the reviewer is very welcome to watch from the link in the paper, see ref 23). Evaluative approaches to prior iterations of the UIPC course have been published elsewhere:

Jowsey, T., Yu, T. C. W., Ganeshanantham, G., Torrie, J., Merry, A. F., Bagg, W., ... & Weller, J. (2018). Ward calls not so scary for medical students after interprofessional simulation course: a mixed-methods cohort evaluation study. BMJ Simulation and Technology Enhanced Learning, 4(3), 133-140.

Performativity theory is appropriate for use in this study but I would like the authors to explain why they selected this framework - is it novel? Previously used by these investigators? Does it have a particular strength or nuance that helps this study? I found the sentence on page 10 lines 253-254 rather opaque - please clarify what you intend by theatirical and improvisational components. An example would be helpful.

Changes made. We have completely reworked this section of the manuscript to make it easier to understand why we chose performativity theory and to make that part of the manuscript easier to understand for readers who are not performance theorists or ethnographers. [lines 254-258 and 262-267]

Similarly, please expand and illustrate your definition of risk. There are many ways to talk about and conceptualize risk, but I am unsure of the specifics of what you are using. 

Change made to define risk [lines 283-288]: In this paper, we are conceptualising risk in terms of the way a person perceives their personal exposure or the exposure of their patient to danger, harm, or loss (including loss of face). In the case of personal risk, this may involve risk of being seen to make an error and being judged socially – by teachers and peers – on performance.

Also, what do you mean by "fleeting moments of decision making (p 11 line 272)? please explain and illustrate in the context of the UIPCW. 

Text changed to clarify: …and often rapid moments of decision making. In the simulation setting, people are often required to quickly make decisions and act on them, where risks are either embraced, shied away from, ignored, or abandoned. 

The three figures mentioned on page 13 were absent from my version of the manuscript.

Sorry, they are photos of students during simulations, they were uploaded with the submission. We are not sure why the reviewer couldn’t see them. We hope that if the reviewer does see them and the film they will get a richer understanding of our work. Many thanks to reviewer two for their time in reading and reviewing with these helpful comments.

We really appreciate the insights from these two reviewers that have enabled us to polish the manuscript. Much appreciated. Kind wishes from the authors,

---

## [Decision Letter · Decision Letter 1]

30 Jun 2020

Performativity, identity formation and professionalism: ethnographic research to explore student experiences of clinical simulation training

PONE-D-19-27048R1

Dear Dr. Jowsey,

We’re pleased to inform you that your manuscript has been judged scientifically suitable for publication and will be formally accepted for publication once it meets all outstanding technical requirements.

Kind regards,

Janhavi Ajit Vaingankar

Academic Editor

PLOS ONE

Reviewers' comments:

Reviewer's Responses to Questions

**Comments to the Author**

1. If the authors have adequately addressed your comments raised in a previous round of review and you feel that this manuscript is now acceptable for publication, you may indicate that here to bypass the “Comments to the Author” section, enter your conflict of interest statement in the “Confidential to Editor” section, and submit your "Accept" recommendation.

Reviewer #1: All comments have been addressed

Reviewer #2: All comments have been addressed

2. Is the manuscript technically sound, and do the data support the conclusions?

Reviewer #1: Yes

Reviewer #2: (No Response)

3. Has the statistical analysis been performed appropriately and rigorously? 

Reviewer #1: N/A

Reviewer #2: (No Response)

4. Have the authors made all data underlying the findings in their manuscript fully available?

Reviewer #1: Yes

Reviewer #2: (No Response)

5. Is the manuscript presented in an intelligible fashion and written in standard English?

Reviewer #1: Yes

Reviewer #2: (No Response)

6. Review Comments to the Author

Reviewer #1: I would like to thank the authors for addressing all of the comments raised in my initial review. The use of focused ethnography is highly valuable in healthcare education and practice. I look forward to reading the published article.

Reviewer #2: (No Response)

7. PLOS authors have the option to publish the peer review history of their article (what does this mean?). If published, this will include your full peer review and any attached files.

Reviewer #1: **Yes: **Dr Suzanne Gough

Reviewer #2: No

---

## [Editor Report · Acceptance letter]

6 Jul 2020

PONE-D-19-27048R1 

Performativity, identity formation and professionalism: ethnographic research to explore student experiences of clinical simulation training 

Dear Dr. Jowsey:

I'm pleased to inform you that your manuscript has been deemed suitable for publication in PLOS ONE. Congratulations! Your manuscript is now with our production department. 

Kind regards, 

on behalf of

Ms Janhavi Ajit Vaingankar 

Academic Editor

PLOS ONE